# Foot Kinematics of Elite Female 59 kg Weightlifters in the 2018–2021 World Championships

**DOI:** 10.3390/jfmk9040207

**Published:** 2024-10-26

**Authors:** Wei-Cian Yan, Pei-Tzu Lan, Chia-Fang Wu, Wen-Pin Hu

**Affiliations:** Department of Bioinformatics and Medical Engineering, Asia University, Taichung 413305, Taiwan; 111225007@live.asia.edu.tw (W.-C.Y.); 112225001@live.asia.edu.tw (P.-T.L.); 113225003@live.asia.edu.tw (C.-F.W.)

**Keywords:** Olympic weightlifting, motion analysis, foot positioning, athletic performance

## Abstract

Background/Objectives: Research on elite weightlifting performance is crucial for understanding the underlying attributes of efficient techniques. This study aimed to analyze the foot characteristics of elite female weightlifters in the 59 kg category during the snatch. Methods: Publicly available videos from the International Weightlifting Federation World Weightlifting Championships (2018–2021) were analyzed. Excluding the 2020 competition due to the COVID-19 pandemic and more unsuccessful attempts, a total of 20 videos were selected for kinematic analysis using Kenova video analysis software. Variables included the horizontal foot distance in the start and catch phases, horizontal displacement of sideway leg separation, and maximum vertical heel height of each foot. Results: The results revealed small to moderate significant negative correlations between snatch performance and maximum heel height (right: r = −0.28, *p* < 0.05; left: r = −0.332 *p* < 0.01). Snatch performance also demonstrated a small to moderate negative correlation with sideway leg separation and foot distance in the catch phase (r = −0.275, *p* < 0.01; r = −0.467, *p* < 0.01, respectively). Maximum heel height exhibited a very strong positive correlation between feet (r = 0.853, *p* < 0.01). Conclusions: A relatively narrower stance was found to be more beneficial for elite weightlifter performance. Strong coordination suggests advanced movement strategies in this complex lift. These findings contribute to the existing knowledge on weightlifting techniques and offer valuable insights for athletes and coaches seeking to improve performance in competitive environments.

## 1. Introduction

Olympic weightlifting stands out as a sport that requires diverse physical attributes such as strength, speed, power, coordination, and technique [1]. Athletes in this discipline compete by lifting weights from the ground to an overhead position in two events: the snatch and the clean and jerk. This task demands not only immense physical prowess but also precise technical execution. Historically, weightlifting was dominated by male athletes. However, the gradual inclusion of female participants began in the late 1800s, although opportunities were limited for women. The 1980s marked a significant turning point, when dedicated female-only competitions were established. A major milestone was achieved in 1987 with the inaugural Women’s World Weightlifting Championship, solidifying women’s presence in the sport [2]. In 2000, women’s weightlifting was officially included in the Olympic Games, further cementing their equal status within the sport on the global stage [3]. Over the years, weightlifting has seen increasing popularity worldwide, and more women have entered the sport, pushing the boundaries of performance and participation. However, despite the sport’s growth, research focusing on female weightlifters remains limited compared to that on male athletes [3,4]. To gain a comprehensive understanding of the physiological and performance aspects that are unique to women in weightlifting, it is crucial to prioritize research that specifically addresses the needs and experiences of women in this sport. This focus will help in developing tailored training programs and enhancing performance outcomes for female weightlifters.

Weightlifting movements have been shown to provide effective improvements in power, speed, and strength compared with other forms of training [5,6,7]. The triple extension of the hip, knee, and ankle joints during the second pull phase is a characteristic feature of weightlifting movements, which has the greatest transference to athletic performance [8,9]. Most studies analyzing the technique of the snatch have primarily focused on the kinematics of the joints [10,11,12]. Optimizing the synchronization of the entire kinematic chain in weightlifting is essential for achieving peak performance, as it requires the coordinated activation of all body segments [12]. Moreover, the feet play a crucial role in this process, serving as a dynamic base of support throughout the entire lift and significantly contributing to lift efficiency and stability [8]. AkkuS [11] analyzed the angular kinematics of the lower limb joints during the snatch, revealing that the maximum internal rotation angle of the ankle extension was 149.06 ± 5.32° during the lift. Similarly, Korkmaz and Harbili [12] found that the maximum ankle extension angle was 138.17 ± 5.49°. However, few studies have combined variables that consider foot positioning in their research. A study demonstrated that no significant differences were observed in the plantar pressure distribution between female national-level competitive weightlifters and female physical education students during standing still and walking [13]. However, while these findings suggest similarities in static and basic dynamic conditions, they do not account for the more complex biomechanical demands of weightlifting movements. Assessing foot positioning during lifts remains critical for understanding the role of the feet in maintaining balance, generating force, and ensuring stability, especially during high-intensity lifts where variations in technique can significantly impact performance. Consequently, it is necessary to explore foot biomechanics during dynamic lifting tasks to provide insights into how athletes can optimize their technique and enhance performance outcomes.

In sports science, the biomechanical analysis of athletes’ movements offers valuable insights for coaches and athletes. Camera recording is a widely used technique in biomechanical analysis. To gather precise and comprehensive movement data, athletes frequently undergo extensive testing in biomechanical laboratories [10,14,15,16]. These facilities are equipped with sophisticated and costly technology, including multi-camera 3D motion capture systems and force platforms [16]. In a previous study, researchers employed a three-camera system in a laboratory setting to construct a two-dimensional spatial model using 14 reflective markers placed on bony landmarks [10]. This model was used to reconstruct the geometry of the weightlifters in the anterior frontal plane and the right and left sagittal planes. Moreover, camera recording can be used in field settings. For instance, Akkuş [11] used two digital cameras positioned diagonally above the platform at a 9 m distance during the 2010 World Weightlifting Championship, creating a 45-degree angle with the sagittal plane of female weightlifters. The kinematic analysis revealed that the angular displacements and velocities of the ankle, knee, and hip joints were greater during the second pull compared to the first pull. The average knee angle was 66.32 ± 12.96° at the start, 134.14 ± 4.73° after the first pull, 11.02 ± 5.32° during the transition phase, and 159.09 ± 2.74° at the end of the second pull. Cunanan et al. [17] also used a GoPro digital video camera to capture the successful snatch attempts in the 2015 World and 2017 Pan-American Weightlifting Championships. However, practical challenges such as limited space and equipment constraints can hinder the implementation of comprehensive camera setups in real-world environments. Recent studies have explored the feasibility of using publicly available videos for motion analysis. For example, Kong et al. [18] demonstrated the potential of performing meaningful movement analysis from publicly available videos using free software to analyze acrobatic sports. Similarly, Luxem et al. [19] highlighted the effectiveness of open-source tools in behavioral video analysis, emphasizing their scalability and reproducibility. This approach allows researchers to generate valuable kinematic data without the need for expensive equipment or controlled laboratory settings. Meanwhile, the use of publicly accessible videos has allowed for the analysis of real-time performance at the highest competitive level.

Therefore, the primary objective of this study is to examine the kinematic characteristics of foot positioning during the snatch in world-class female weightlifters using public available videos. The sample population consisted of elite weightlifters who secured top three finishes in the 59 kg category at the World Weightlifting Championships from 2018 to 2021. Our hypothesis posits that a kinematic evaluation of the snatch in these elite female athletes will reveal noteworthy insights into foot-related variables, a potentially understudied area within the context of weightlifting performance.

## 2. Materials and Methods

### 2.1. Experimental Approach to the Problem

Publicly available videos from the World Weightlifting Championships on YouTube were retrieved for kinematics analysis in this study. To ensure the accuracy and minimize potential biases caused by the position and angle of the recording device, we only analyzed video recordings captured from the anterior frontal plane. Therefore, the World Weightlifting Championship from 2018 to 2021, held in Ashgabat, Turkmenistan (2018), Pattaya, Thailand (2019), and Tashkent, Uzbekistan (2021), were selected. It is important to note that the 2020 World Weightlifting Championship was canceled due to the global COVID-19 pandemic. Additionally, we collected successful snatch attempts by the top three female weightlifters in the 59 kg category, which was selected for analysis, because it was considered by meet officials to be one of the elite categories with the best potential for setting a world record. The study was approved by the Institutional Review Board of the Jen-Ai Hospital (ID: 202300008B0).

### 2.2. Procedures and Video Analysis

The analysis was supposed to comprise a total of 27 individual attempts from the top three female weightlifters in the 59 kg category. However, one of the lifters at the 2018 World Weightlifting Championship was not recorded due to her assignment to group B. In the World Weightlifting Championships, the athletes are divided into groups based on their qualifying total or other criteria. Group B typically refers to the group of athletes who have qualified for the competition but did not place high enough in their qualifying event to be included in group A [20]. Both group A and group B athletes compete in their respective groups, and group B attempts are often not broadcast. Consequently, the final sample included 20 successful attempts from a total of 8 athletes (Table 1). All videos were extracted from YouTube with a resolution of 640 × 360 pixels at a frame rate of 30 Hz using OBS Studio 29.1.3 software. This method allowed for a consistent and reliable collection of data from the video recordings.

The Kinovea software (Version 0.8.26, Kinovea open-source project, available for download at www.kinovea.org) was employed to measure the variables derived from the videos. Kenova is a free, open-source 2D motion analysis software licensed under GPLv2. It is a widely used 2D motion analysis software in the field of sports kinematics, with demonstrated high accuracy in measuring distances and displacements [16]. When choosing biomechanical variables for analysis, it is essential to consider factors such as the recording rate, camera positioning, and scaling factors. Distance measurements can be readily acquired if the key phases (e.g., start phase and catch phase) are clearly visible in the video footage.

Firstly, the length of the bar was recorded as 201 cm to calibrate the horizontal axis in the video, in accordance with the standard dimensions of the women’s Olympic bar. We also conducted visual inspections for distortions and compared the image with known vertical lines to minimize errors. Second, the horizontal foot distance was measured during the start and catch phases of the snatch (Figure 1A,C,D,F). To ensure consistency, all measurements were taken from the outermost point of the shoes. Additionally, the lateral displacement of the feet was calculated by subtracting the start phase distance from the catch phase distance. Finally, the maximum vertical heel height prior to toe-off during the second pull was measured (Figure 1B,E). Video analyses were conducted by three raters under the supervision of a PhD in sports biomechanics with a competitive background in Olympic weightlifting. To ensure reliability, each video was analyzed independently by all raters.

### 2.3. Statistical Analysis

A statistical significance level of *p* < 0.05 was utilized for all tests in this study. All statistical analyses were conducted using SPSS version 25.0 (IBM, Armonk, NY, USA). To assess the inter-rater reliability, each of the three raters independently analyzed a sub-sample of 20 videos. The intraclass correlation coefficients (ICCs) for the variables were calculated, and the standard error of measurement (SEM) for each variable was computed. The Shapiro–Wilk test was employed to assess the normality of data. Pearson’s correlation coefficient was used to explore the relationships between variables. The magnitudes of the effect for the correlations were interpreted as follows: trivial (≤0.09), small (0.10–0.29), moderate (0.30–0.49), large (0.50–0.69), very large (0.70–0.89), and nearly perfect (≥0.9) [22]. For each variable, the mean and standard deviation (SD) were used to provide a descriptive summary.

## 3. Results

The inter-rater reliability among the three raters was found to be excellent for the measurement of foot distance during both the start phase (ICC = 0.991; SEM = 0.64 cm) and the catch phase (ICC = 0.997; SEM = 0.64 cm), as well as for the sideway displacement (ICC = 0.992; SEM = 0.78 cm). Similarly, the maximum vertical heel height of both feet demonstrated good reliability (right foot: ICC = 0.79, SEM = 0.27 cm; left foot: ICC = 0.906, SEM = 0.30 cm). The snatch exhibited significant small to moderate negative correlations with the heel height of both the right and left foot (r = −0.28, *p* < 0.05; r = −0.332, *p* < 0.01, respectively, Figure 2).

Table 2 illustrates significant correlations between the snatch and various variables. The sideway displacement and the distance between the feet in the catch phase displayed significant moderate to small negative correlations with the snatch (r = −0.275, *p* < 0.01; r = −0.467, *p* < 0.01, respectively). Furthermore, the distance between the feet in the start phase showed a small and positive correlation with the distance between the feet in the catch phase (r = 0.262, *p* < 0.05). On the other hand, the results revealed a significant and large negative correlation between the distance between the feet in the start phase and the sideway displacement of the feet (r = −0.603, *p* < 0.01). The distance between the feet in the catch phase exhibited significantly positive correlations with all variables except for the snatch and the distance between the feet in the start phase. Additionally, a significant and very large positive correlation was observed with the heel heights of the feet (r = 0.853, *p* < 0.01). 

## 4. Discussion

This study used Olympic weightlifting as a case study to demonstrate how movement analysis can be conducted using publicly available videos and free software, providing valuable insights into high-performance sports. Assessing foot positioning and plantar pressure during weightlifting movements remains crucial for understanding the feet’s role in maintaining balance, generating force, and ensuring stability. Proper foot placement is essential for optimal load transfer and injury prevention, especially during high-intensity lifts, where subtle variations in technique can significantly impact the performance. In this study, we conducted an investigation into various variables of foot kinematics of the snatch by the top three female weightlifters in the 59 kg category at the World Weightlifting Championships in the period from 2018 to 2021. The variables included (1) the distance between the feet in the start and catch phase, (2) the sideway displacement of the feet, and (3) the maximum vertical height of both the heels. The findings revealed that, while a wider stance did not significantly enhance performance, the strong coordination observed suggests the use of advanced movement strategies in this complex lift. The videos analyzed were of elite athletes performing in high-level competitions, providing greater ecological validity compared to laboratory biomechanical tests [23]. Despite the videos being recorded with varying camera settings and qualities, the proposed data analysis approach demonstrates good to excellent inter-rater reliability for measuring the distance between the feet, sideway displacement, and maximum vertical heel height during the snatch. This study illustrates that publicly available resources, such as YouTube videos, can be effectively utilized to enhance our understanding of sports movements at a low cost.

Previous studies have investigated various biomechanical parameters of snatch lift performance during successful lifts, utilizing both two-dimensional and three-dimensional kinematic analysis techniques [4,8,16,19,23]. Research has shown that elite weightlifters exhibit similar characteristics in their limb and barbell movements during the lift [1]. The athletes in this study belonged to the 59 kg category, and their heights showed minimal deviation (Table 1). To the best of our knowledge, the variables investigated in this study have not been extensively explored in previous analyses. Chavda et al. [24] have provided a comprehensive definition of the stable and variable components within the technical weightlifting model. Among these, they identified foot position as a variable component. Their research suggests that the width of an athlete’s foot stance can vary based on genetic factors and the anatomical positioning of the femoral head within the acetabulum. This implies that during the start phase of a lift, the foot width should ideally resemble the stance used in a vertical jump to optimize performance. In our study, the distance between the feet in the start phase was 42.24 ± 4.90 cm. However, future studies should incorporate anthropometric assessments for a more comprehensive comparison with previous research.

Previous studies have shown that increasing the weight of the barbell significantly impacts the kinematic factors of the snatch [4,25]. One study found that as the weight of the barbell increased, the maximum vertical displacement of the barbell decreased [1]. This reduction in vertical displacement implies greater restrictions on ankle extension, a finding that is consistent with the results of our investigation. Specifically, we observed small to moderate negative correlations between the weight lifted during the snatch and heel height, suggesting that as the barbell weight increases, the height of the heel lift decreases (Figure 2). This relationship supports the idea that heavier loads limit an athlete’s ability to fully extend at the ankles, thereby reducing the heel elevation. Additionally, it has been observed that heavier barbells lead to changes in the timing and coordination of the lift, requiring athletes to adapt their technique to maintain performance [4,25]. The consistent foot movements observed in elite-level performances further validate the accuracy and reliability of utilizing publicly available videos for kinematic analysis. All videos analyzed in this study were captured during high-level competitions featuring elite female weightlifters, meaning that the weights they lifted were close to their one-repetition maximum (1 RM). This context provides a realistic and challenging scenario for analyzing the biomechanics of the snatch, ensuring that the findings are applicable to actual competitive conditions.

Our results revealed a very large positive correlation between the heel heights of the right and left foot (Table 2), indicating a high degree of bilateral symmetry in foot movement during the snatch in elite weightlifters. This finding was corroborated by Mira, who reported bilateral symmetry in the articular kinematics during the snatch [10]. Bilateral symmetry in weightlifting is essential for balanced muscle development and injury prevention. Studies have shown that asymmetries in movement patterns can lead to compensatory mechanisms, which may increase the likelihood of overuse injuries [26]. This balance is also further supported by a study that demonstrated no significant differences in peak moment powers between the left and right lower extremities during the snatch, reinforcing the notion of symmetrical lower body mechanics [15]. In our study, the consistent heel heights observed suggest that athletes maintain a stable and balanced stance throughout the lift, which is crucial for optimal performance. Additionally, maintaining bilateral symmetry can enhance proprioception and coordination, allowing athletes to execute lifts with greater precision and control.

The role of the feet in snatch performance is fundamental, as they provide a dynamic and stable base throughout the lift, ensuring proper balance and force transmission [27]. The snatch begins from a squatting position, followed by a dynamic and explosive motion to elevate the barbell overhead in one fluid movement [28]. During the catch phase, many athletes adopt a split stance by widening their feet to create a stable base for receiving the barbell. Optimal foot displacement in this phase is essential not only for generating maximal force but also for maintaining balance and improving the overall movement efficiency. Proper foot positioning is crucial for avoiding energy leaks and enhancing the lifter’s ability to handle maximal loads safely [11]. The results of this study demonstrated a significant positive correlation between the distance between the feet in the catch phase and both the initial foot placement in the start phase and the overall foot displacement (Table 2). These findings emphasize the importance of foot displacement in the snatch, particularly among elite female weightlifters, where proper foot positioning plays a key role in successful lift execution. Previous research has also shown that the distance between the feet in the catch phase typically exceeds the shoulder width, further underscoring the necessity of widening the stance to improve performance [10].

Our analysis identified a significant negative correlation between the distance between the feet in the catch phase and the weight lifted during the snatch (Table 2). This finding aligns with earlier research indicating that an overly wide foot placement can reduce the efficiency of the force transfer and potentially compromise the lift execution, particularly under heavier loads [10]. This suggests that while some degree of foot separation is essential for balance and stability, an excessively wide stance may be detrimental. This finding could be attributed to the biomechanical limitations imposed by the skeletal structure and function of the human body [29]. Specifically, the hips play an important role in determining the range of motion and stability during weightlifting movements [30]; however, the degree of external rotation and abduction that occurs during the catch phase can be constrained by individual anatomical differences [31]. When the feet are placed too far apart, the lifter may experience diminished hip mobility and reduced capacity to produce force efficiently. An overly wide stance may increase the risk of instability in the lower extremities, as the athlete’s center of mass becomes harder to control during the dynamic phases of the lift [32]. Furthermore, excessive lateral foot displacement can place undue stress on the knee joints and other lower body structures, increasing the potential for injury [24].

The findings of this study suggested that a relatively narrower stance was found to be more beneficial for elite weightlifter performance. Excessive widening of the stance may hinder weightlifting performance, emphasizing the need for a balanced approach to foot positioning that optimizes both stability and force generation. Achieving an optimal balance between foot positioning, stability, and force production is essential for maximizing performance in the snatch, particularly at elite levels where small adjustments can significantly impact the weight lifted and the overall success of the lift. By optimizing the foot displacement, lifters can improve their ability to balance the barbell, maintain stability, and complete the snatch with greater efficiency and success.

### Limitations

This study faced several limitations. First, as the analysis relied on publicly available videos, we only selected match sessions with anterior frontal images to minimize errors associated with varying camera angles. Also, the use of 30 frames per second (fps) for video analysis may have introduced some limitations. While 30 fps is sufficient for capturing many human movements [14], faster frame rates (e.g., 50–60 fps) can provide more precise data for analyzing rapid movements. These restrictions reduced the sample size and highlight the challenges of using publicly available footage, where not all performances may be captured or accessible for detailed analysis. Second, this study specifically targeted elite female weightlifters. Given the differences in strength and weight lifted between male and female athletes, the correlations observed in this study may not directly apply to male lifters [33]. Additionally, focusing on a specific bodyweight category may limit the generalizability of the findings. Future research should explore the potential differences in foot kinematics and snatch performance between genders and across various weight categories to provide a broader understanding of these variables. Third, the physical dimensions of the athletes, such as limb length, foot size, and hip width, were not measured in this study. These factors could influence the foot positioning, movement mechanics, and overall kinematics during the snatch. Their exclusion may limit the depth and precision of the analysis. To address this gap, future studies should incorporate detailed anthropometric data to gain a more comprehensive understanding of how physical characteristics impact the foot positioning and overall performance in elite weightlifting.

## 5. Conclusions

The use of publicly available videos and open-source software provides a cost-effective and accessible means to conduct such detailed motion analyses [19], expanding the scope of research and practical applications in sports science. In weightlifting, incorporating foot positioning variables into the analysis could enhance the precision of technique assessments and contribute to the development of more effective training programs. This study analyzed the foot positioning of elite female 59 kg weightlifters, suggesting that a relatively narrower stance may be more beneficial for elite weightlifter performance. Furthermore, strong coordination suggests advanced movement strategies in this complex lift. Understanding these correlations can help coaches and athletes optimize performance by focusing on key aspects of technique. In conclusion, the findings of this study offer valuable insights into foot positioning in female weightlifters, providing practical implications for refining training strategies and enhancing performance in both practice and competition settings.

## Figures and Tables

**Figure 1 jfmk-09-00207-f001:**
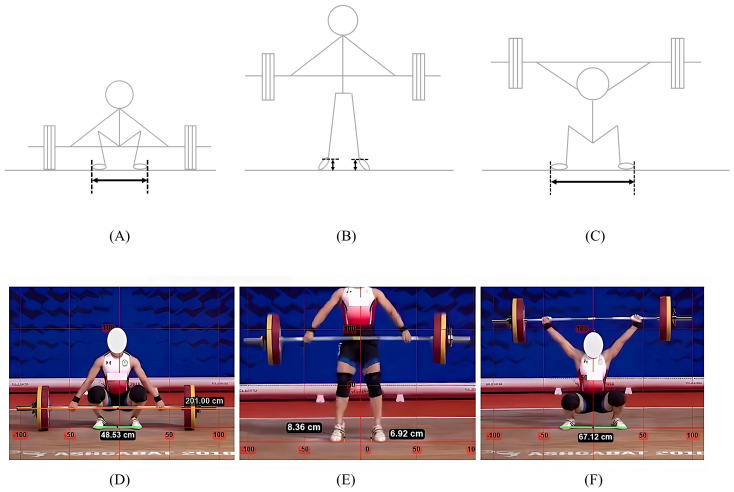
Definition and investigation of variables during the snatch: (**A**) the distance between the feet in the start phase; (**B**) the maximum height of the heels; (**C**) the distance between the feet in the catch phase. (**D**–**F**) Variables were investigated using Kinovea software through the analysis of publicly available videos [21].

**Figure 2 jfmk-09-00207-f002:**
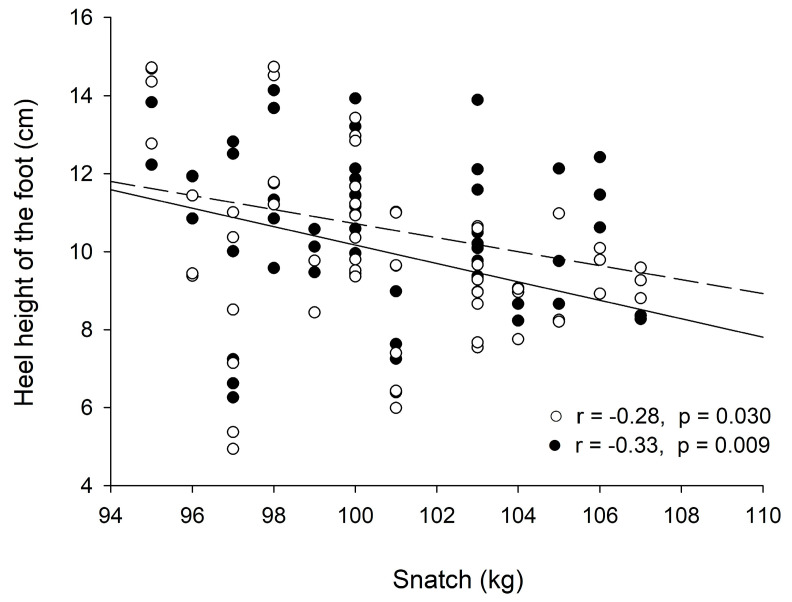
Correlation between the snatch and the maximum heel height of each foot in 59 kg category elite female weightlifters. (●) indicates the right and (○) indicates the left foot; (──) represents the regression line of the right foot and (-----) represents the left.

**Table 1 jfmk-09-00207-t001:** The characteristics of elite female weightlifters in the 59 kg category.

Athletes	Age (Year)	Body Weight (kg)	Height (cm)	Best Result (kg)
A	25	58.07	155	105
B	20	58.70	160	103
C	26	58.95	159	107
D	26	58.55	155	106
E	25	58.85	157	101
F	21	58.90	155	101
G	28	58.80	155	100
H	25	58.80	157	100
Mean ± SD	24.50 ± 2.50	58.70 ± 0.27	156.63 ± 1.87	102.88 ± 2.62

**Table 2 jfmk-09-00207-t002:** Correlation coefficients between variables in 59 kg category elite female weightlifters.

	Lifted Weight	Foot Distance (Start Phase)	Foot Distance (Catch Phase)	Sideway Displacement	Maximum Heel Height (Right)	Maximum Heel Height (Left)
Lifted weight		−0.135	−0.467 **	−0.275 *	−0.280 *	−0.332 **
Foot distance (start phase)	−0.135		0.262 *	−0.603 **	0.084	0.194
Foot distance (catch phase)	−0.467 **	0.262 *		0.612 **	0.394 **	0.486 **
Sideway displacement	−0.275 *	−0.603 **	0.612 **		0.257 *	0.243
Maximum heel height (right)	−0.280 *	0.084	0.394 **	0.257 *		0.853 **
Maximum heel height (left)	−0.332 **	0.194	0.486 **	0.243	0.853 **	

Note: Significant correlations between variables at * *p* < 0.05 and ** *p* < 0.01.

## Data Availability

The raw data supporting the conclusions of this article will be made available by the authors on request.

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
