# Peer review of "Foot Kinematics of Elite Female 59 kg Weightlifters in the 2018–2021 World Championships"

_jfmk, 2024, doi:10.3390/jfmk9040207_

Round 1
Reviewer 1 Report
Comments and Suggestions for Authors
(1) I think that this study's approach and findings are interesting. These findings will contribute to the education and practice of weightlifting. In addition, the findings of this study can be applied for other than sports areas such as occupational manual handling. Good job.
(2) What is the most novel aspect of this study? In my opinion, movement analysis using Kinovea has been conducted in previous various studies. On the other hand, I think that findings about the relationship between snatch (weight lifting) technique and foot placement will be novel in this paper. Please highlight the novelty of this study in the abstract and introduction.
(3) The authors could describe the procedure in the manuscript well. I think that the carburation method using the standard length of a bar is interesting and useful.
(4) Figure 1 includes screenshots of Olympic videos obtained from YouTube. Are there any copyright issues?
(5) The authors could discuss each trend of experimental results by comparison with previous studies.
(6) This study focuses on successful snatch. In future works, these kinematics will be compared with failed or beginner's techniques?
(7) I think it interesting that "while some degree of foot separation is essential for balance and stability, an excessively wide stance may be detrimental" in your manuscript. These findings and recommendations are beneficial knowledge for me.
(8) Kinovea has a function for object tracking. In future works, will you investigate the relationships between foot placement and dynamic parameters such as knee movement or timing of changing feet stance?
Author Response
Comments 1: I think that this study's approach and findings are interesting. These findings will contribute to the education and practice of weightlifting. In addition, the findings of this study can be applied for other than sports areas such as occupational manual handling. Good job.
Response 1: Thank you very much for your encouraging comments.
Comments 2: What is the most novel aspect of this study? In my opinion, movement analysis using Kinovea has been conducted in previous various studies. On the other hand, I think that findings about the relationship between snatch (weight lifting) technique and foot placement will be novel in this paper. Please highlight the novelty of this study in the abstract and introduction.
Response 2: Thank you for your comments. Yes, the primary novelty of this study lies in its in-depth analysis of foot displacement in elite female weightlifters during the snatch, using a publicly available dataset. While movement analysis using Kinovea has been conducted in previous studies, this study specifically examines foot characteristics, which may be a less explored area within the context of weightlifting performance. We have amended the abstract and the final paragraph of the Introduction.
Comments 3: The authors could describe the procedure in the manuscript well. I think that the carburation method using the standard length of a bar is interesting and useful.
Response 3: Thank you very much for your encouraging comments.
Comments 4: Figure 1 includes screenshots of Olympic videos obtained from YouTube. Are there any copyright issues?
Response 4: Thank you for your comments. All publicly available videos used in this study were from the World Weightlifting Championships on YouTube, not the Olympic Games. The Olympic Broadcasting Rights may not be considered in this study. However, there may be potential copyright concerns associated with using screenshots from publicly available videos in the manuscript. Therefore, we have covered the athlete’s face and add the citation the source of the video in the figure legend of figure 1. On the other hand, in our review of the literatures, a related study that also analyzed publicly available videos and stated that there were no copyright concerns regarding the inclusion of these videos screenshots in their paper (specifically, it is #19 of the references in updated manuscript).
Comments 5: The authors could discuss each trend of experimental results by comparison with previous studies.
Response 5: Thank you for your suggestion. To our knowledge, this study is the first to investigate foot displacement during the snatch. However, we have expanded the discussion to compare the trends in our results with those of previous studies.
Comments 6: This study focuses on successful snatch. In future works, these kinematics will be compared with failed or beginner's techniques?
Response 6: Thank you for your comments. Yes, the primary objective of this study is to identify the foot characteristics of elite female weightlifters during the snatch. Therefore, unsuccessful attempts were not included in this study. However, we appreciate your valuable suggestions and will consider them for future research.
Comments 7: I think it interesting that "while some degree of foot separation is essential for balance and stability, an excessively wide stance may be detrimental" in your manuscript. These findings and recommendations are beneficial knowledge for me.
Response 7: Thank you very much for your encouraging comments.
Comments 8: Kinovea has a function for object tracking. In future works, will you investigate the relationships between foot placement and dynamic parameters such as knee movement or timing of changing feet stance?
Response 8: Thank you very much for your comments. We would like to include more kinematic characteristics in future work.

Reviewer 2 Report
Comments and Suggestions for Authors
The authors provided a well written manuscript that fits within the aims of the Journal. However, there are topics that need to be addressed.
1. L74-97. Further elaboration is required on the published biomechanical research on weightlifting techniques.
2. The rationale to examine the foot positioning in female 59-kg weighlifters needs to be highlighted. So far, no information about the possible differences in this specific group is provided, nor about the importance of the examined variables in the optimization of weightlifting performance.
3. No clear hypothesis is provided: what did the researchers expect to find?
4. L116-118: provide reference for this statement.
5. Clarify 'Group B' in L126.
6. Is the sampling frequency (30fps) [L128] adequate to examine the kinematics of the examined parameters, as the motion is rather fast?
7. L138-151: For 2D analysis, important elements is the verticality of the camera and the absence of an angle of the camera optical axis with the plane of interest. How did the authors checked if these two requirements were satisfied? In addition, calibration was conducted for the horizontal axis but the measures were done for the vertical axis. If these requirements were not satisfied, errors might occur for the reported findings.
8. Given that the above-mentioned requirements were satisfied, then the anthropometrics mentioned in the limitations (i.e. lower limb segment lenghts, hips width [L307-308]) could have been measured and thus to provide further information that could add to the Discussion.
9. Conclusions: Practical suggestions for coaches and practitioners are expected.
10. References: Type citations as in the Journal's Author Guidelines. Also: delete #31 (L398].
Comments on the Quality of English LanguageMinor editing of English language required (e.g., synchronization instead of synchrony in L55).
Author Response
<Reviewer 2>
General Comments: The authors provided a well written manuscript that fits within the aims of the Journal. However, there are topics that need to be addressed.
Response: Thank you very much for your encouraging comments.
Comments 1: L74-97. Further elaboration is required on the published biomechanical research on weightlifting techniques.
Response 1: Thank you for your comment. We have expanded the third paragraph in Introduction.
Comments 2: The rationale to examine the foot positioning in female 59-kg weighlifters needs to be highlighted. So far, no information about the possible differences in this specific group is provided, nor about the importance of the examined variables in the optimization of weightlifting performance.
Response 2: Thank you for your suggestion. The female 59-kg category was selected for analysis because it was considered by meet officials to be one of the elite categories with the best potential for setting a world record. This information has been updated in section 2.1.
Comments 3: No clear hypothesis is provided: what did the researchers expect to find?
Response 3: Thank you for your comment. To our knowledge, this study is the first to investigate foot displacement and heel height during the snatch. Therefore, we hypothesize that kinematic evaluation of the snatch in these elite female weightlifters will reveal noteworthy insights into foot-related variables, a potentially understudied area within the context of weightlifting performance.
Comments 4: L116-118: provide reference for this statement.
Response 4: Thank you for your comment. We have amended the sentence to “The female 59-kg category was selected for analysis because it was considered by meet officials to be one of the elite categories with the best potential for setting a world record”.
Comments 5: Clarify 'Group B' in L126.
Response 5: Thank you for your comment. We have included more information about 'Group B' in section 2.2 and also cited a new reference about this classification system from International Weightlifting Federation (specifically, it is study 21).
Comments 6: Is the sampling frequency (30fps) [L128] adequate to examine the kinematics of the examined parameters, as the motion is rather fast?
Response 6: Thank you for your comment. In our review of the literatures, while 50-60 fps is commonly used in motion analysis, one study has demonstrated the effectiveness of 30 fps (specifically, it is study 15). However, we acknowledge that 30 fps may have limitations, and this aspect has been added in the discussion of research limitations.
Comments 7: L138-151: For 2D analysis, important elements is the verticality of the camera and the absence of an angle of the camera optical axis with the plane of interest. How did the authors checked if these two requirements were satisfied? In addition, calibration was conducted for the horizontal axis but the measures were done for the vertical axis. If these requirements were not satisfied, errors might occur for the reported findings.
Response 7: Thank you for your comment. It is crucial to assess camera angle and verticality when analyzing public videos. In this study, we exclusively selected match sessions captured from an anterior frontal perspective to minimize errors arising from varying camera angles (which means only one camera was used). Since we did not have direct control over the camera setup during the competition, we not only calibrated the horizontal axis but also checked for visible distortions and compared the image with known vertical lines to limit errors in this study. We have expanded the information in the third paragraph of section 2.1.
Comments 8: Given that the above-mentioned requirements were satisfied, then the anthropometrics mentioned in the limitations (i.e. lower limb segment lenghts, hips width [L307-308]) could have been measured and thus to provide further information that could add to the Discussion.
Response 8: Thank you for your suggestion. The primary objective of this study is to identify the foot characteristics of elite female weightlifters during the snatch. We will include more kinematic characteristics (i.e. lower limb segment lengths, hips width) in future work.
Comments 9: Conclusions: Practical suggestions for coaches and practitioners are expected.
Response 9: Thank you for your comment. We have expanded the practical suggestions in Conclusions.
Comments 10: References: Type citations as in the Journal's Author Guidelines. Also: delete #31 (L398].
Response 10: Thank you for your correction. We have deleted reference #31 in the manuscript.

Round 2
Reviewer 2 Report
Comments and Suggestions for Authors
In the resubmitted version of the manuscript, the author(s) addressed the comments raised on the initial round of reviewing. However, some topics need to be further processed.
1. Although additional information about the biomechanics of weightlifting was added, no specific information is provided for the kinematics aquired from the frontal plane and their impact on performance.
2. Given the missing information about the frontal plane kinematics, the hypothesis needs to be clarified. For example, would the foot-related variables be symmetrical/larger etc compared to findings from previous studies?
3. official's opinion: was it evidence based?
Author Response
General Comments: In the resubmitted version of the manuscript, the author(s) addressed the comments raised on the initial round of reviewing. However, some topics need to be further processed.
Response: Thank you and we appreciate your valuable suggestions and comments.
Comments 1: Although additional information about the biomechanics of weightlifting was added, no specific information is provided for the kinematics aquired from the frontal plane and their impact on performance.
Response 1: Thank you for your comment. Although this study is the first to investigate foot displacement and heel height during the snatch, we have expanded the information about the frontal plane kinematics and added the comparison with a previous study (specifically, it is study #25) in the second paragraph of the Discussion.
Comments 2: Given the missing information about the frontal plane kinematics, the hypothesis needs to be clarified. For example, would the foot-related variables be symmetrical/larger etc compared to findings from previous studies?
Response 2: Thank you for your comment. We have expanded the information about the kinematics in the frontal plane in the second paragraph of the Discussion.
Comments 3: official's opinion: was it evidence based?
Response 3: Thank you for your comment. The top one female weightlifter in our country competes in the 59-kg category and has previously held the world record. Therefore, the 59-kg category was initially selected for analysis due to its potential for record-breaking performances. Future research will consider a more comprehensive analysis across all weight categories to provide a more objective evaluation. We also added this information in the limitations of the discussion.
